# Effectiveness of influenza vaccines in adults with chronic liver disease: a systematic review and meta-analysis

Suvi Härmälä,[1] Constantinos A Parisinos,[1] Laura Shallcross,[1] Alastair O'Brien,[2] Andrew Hayward[3]

¹Institute of Health Informatics, University College London, London, UK
²Division of Medicine, University College London, London, UK
³Institute of Epidemiology and Health Care, University College London, London, UK

**Correspondence to**
Suvi Härmälä;
suvi.harmala.14@ucl.ac.uk

## ABSTRACT

**Objectives** Patients with liver disease frequently require hospitalisation with infection often the trigger. Influenza vaccination is an effective infection prevention strategy in healthy and elderly but is often perceived less beneficial in patients with liver disease. We investigated whether influenza vaccination triggered serological response and prevented hospitalisation and death in liver disease.

**Design** Systematic review and meta-analysis.

**Data sources** MEDLINE, EMBASE, PubMed and CENTRAL up to January 2019.

**Eligibility criteria** Randomised or observational studies of the effects of influenza vaccine in adults with liver disease.

**Data extraction and synthesis** Two reviewers screened studies, extracted data and assessed risk of bias and quality of evidence. Primary outcomes were all-cause hospitalisation and mortality. Secondary outcomes were cause-specific hospitalisation and mortality, and serological vaccine response. Random-effects meta-analysis was used to estimate pooled effects of vaccination.

**Results** We found 10 041 unique records, 286 were eligible for full-text review and 12 were included. Most patients had viral liver disease. All studies were of very low quality. Liver patients both with and without cirrhosis mounted an antibody response to influenza vaccination, and vaccination was associated with a reduction in risk of hospital admission from 205/1000 to 149/1000 (risk difference −0.06, 95% CI −0.07 to 0.04) in patients with viral liver disease. Vaccinated patients were 27% less likely to be admitted to hospital compared with unvaccinated patients (risk ratio 0.73, 95% CI 0.66 to 0.80). No effect against all-cause or cause-specific mortality or cause-specific hospitalisation was found.

**Conclusions** The low quantity and quality of the evidence means that the protective vaccine effect may be uncertain. Considering the high risk of serious health outcomes from influenza infection in patients with liver disease and the safety and low cost of vaccination, overall, the potential benefits of seasonal vaccination both to patients and the healthcare systems are likely to outweigh the costs and risks associated with vaccination.

**PROSPERO registration number** CRD42017067277.

## INTRODUCTION

More than 45 million people globally, including >8 million people in the USA and

## Strengths and limitations of this study

▶ The review was conducted following established guidelines and a prospectively specified protocol.
▶ The review focused on all-cause outcomes to reduce the risk of underestimating the vaccine impact that exists due to potential difficulties in identifying and accurately recording the cause of hospitalisation and death in patients with an underlying chronic disease.
▶ To reduce the risk of missing relevant studies and relevant data and to increase the objectivity of the review, two independent reviewers screened the studies, extracted data and assessed the risk of bias and quality of evidence.
▶ Differences in studies included in meta-analyses, publication bias, restriction on the publication language and varying response to queries about data on liver disease subgroups may have affected the effects observed in this review.
▶ Inclusion of non-randomised studies lowers the quality of evidence and limits the conclusions that can be drawn.

Europe, are affected by chronic liver disease.[1] The prevalence of liver disease is increasing, driven by obesity and alcohol consumption.[2]

The progression of liver disease is associated with immune dysregulation[3–6] and complications from common acute infections such as influenza cause significant morbidity and mortality. A twofold increased risk of hospital admission for influenza was observed in patients with liver disease in 19 hospitals in Russia, Turkey, China and Spain during the 2013/2014 season.[7] Similarly, an analysis of data on laboratory-confirmed influenza cases collected from several WHO member states during the 2009 influenza A (H1N1) pandemic found patients with liver disease to have >5-fold increased risk of influenza-related hospitalisation and over 17-fold increased risk of death compared with that of healthy individuals.[8] Influenza infection, while not directly targeting the liver, may

cause collateral liver damage[9] and trigger hepatic decompensation in liver disease.[10]

Influenza vaccine introduces a whole or part of the influenza virion into the body stimulating antibody production against the virus, similar to an actual infection. In healthy adult and elderly populations, seasonal influenza vaccination is an effective way to prevent influenza infection.[11 12] People with liver disease, due to the high risk of severe health complications from influenza infection, are often a target of influenza vaccination policies[13] but randomised trials on vaccine effectiveness in patients with liver disease are lacking as withholding the vaccine may place patients' safety and health at risk. For example, a previous systematic review of randomised trials of influenza vaccine effectiveness in patients with chronic diseases, conducted in 2011,[14] identified only one study in patients with liver disease. Observational studies offer an alternative way of investigating vaccine effectiveness but without randomisation of patients the results are vulnerable to confounding and bias, in particular frailty bias. Frailty bias exist when some or all of the reduction in the risk of health complications may be due to the preferential receipt of vaccine by individuals who are in general more healthy. One solution to minimise the effect of frailty bias is to observe the vaccine effect both within and outside of the influenza season. If frailty bias exist, we would see higher rates of health complications in unvaccinated individuals outside of the influenza season. This would indicate that any protective effect observed within the season may be partially or fully due to the vaccinated individuals being healthier overall and an overestimation of the effect of the vaccine. Finally, in addition to measuring clinical outcomes to understand the effects of the vaccine, the antibody response elicited by the vaccine can be measured and used as a proxy for vaccine effectiveness.

There is uncertainty among primary and secondary care physicians whether influenza vaccines are able to trigger an appropriate antibody response and protect patients with liver disease from health complications,[15] and the vaccination coverage is poor. Just over 55% of the chronic liver disease respondents to the 2016 US National Health Interview Survey[16] had been vaccinated against influenza in the past 12 months and <50% of working-age patients with liver disease registered with primary care practitioners in England had received influenza vaccination during the 2015/2016 season.[17] To inform vaccination policy and to help guide decision-making regarding vaccination, our review aims to provide a systematic synthesis of the available evidence from both observational studies and randomised trials on the effectiveness of influenza vaccines to prevent serious health outcomes, hospitalisation and death, in adults with chronic liver disease. To address uncertainty over the vaccines' ability to trigger antibody response in patients with liver disease, we also assess the effects of influenza vaccines on serological response in adults with chronic liver disease.

## MATERIALS AND METHODS
### Protocol and registration
This systematic review and meta-analysis followed the Preferred Reporting Items for Systematic Review and Meta-Analyses guidelines.[18] The review protocol was specified in advance as part of a wider review plan that also includes a study of the effectiveness of pneumococcal vaccine. The protocol was registered in PROSPERO 13 June 2017 (registration number CRD42017067277) and published after peer review on 16 March 2018.[19] Based on the peer-review feedback, we made minor amendments to the initial protocol (online supplementary table 1).

### Eligibility criteria
Studies published in peer-reviewed journals were eligible for inclusion for this systematic review and meta-analysis if they met the following participants, intervention, comparator, outcome and study design criteria: study type was randomised or non-randomised controlled trial, cohort or case-control study; intervention was an injectable, inactivated whole-virus, split-virus or subunit influenza vaccine; intervention was compared with placebo, alternative intervention or no vaccination (no comparison group was required for serological response); study population included at least 10 adults aged ≥18 years with chronic liver disease of any severity or aetiology (by any classification) per comparison group; outcomes included all-cause hospitalisation or all-cause mortality (or supportive outcomes hospitalisation or mortality due to acute respiratory illness, influenza/influenza-like illness (ILI) or liver disease complications) or serological response to influenza vaccine. We only included studies published in English. Review articles, case reports, cross-sectional studies, animal studies, editorials, clinical guidelines, studies with liver transplant patients only (response to the vaccination and clinical outcomes are likely to be strongly influenced by the immunosuppressive medication rather than the status of liver disease), studies focused on special populations (such as pregnant women, nursing home residents or patients with other chronic diseases), test-negative case-control studies (where both cases and controls were hospitalised for influenza-related illness/ILI/acute respiratory illness, listed in online supplementary table 2) and studies retracted from publication were excluded.

### Information sources and search strategy
We searched the Cochrane Central Register of Controlled Trials, Ovid MEDLINE, Ovid Embase and PubMed (originally up to 16 July 2017 and later updated up to 20 January 2019) for articles containing variations of terms 'influenza', 'influenza', 'seasonal', 'TIV', 'QIV', '3-valent', '4-valent', 'trivalent' or 'quadrivalent' in combination with terms 'vaccine' or 'immunisation' and 'liver disease', 'hepatic disease', 'chronic liver', 'chronic hepatic' or 'cirrhosis'. The search also included medical subject headings 'Influenza Vaccines' (exploded), 'Influenza, human /Prevention and Control' and 'Liver Diseases'

(exploded). The search was filtered by study design, age group of participants and publication type. To capture studies where patients with liver disease may have been included as part of a general population, the search was repeated leaving out the liver disease-specific terms and with the study design filter restricted to trials only. The full search strategy in MEDLINE and the medical subject headings for all databases searched are provided in online supplementary tables 3 and 4. Electronic searches were complemented by manually searching recent influenza vaccine guidelines from Centers for Disease Control and Prevention, European Centre for Disease Control and Prevention and WHO, and the reference lists of the included studies.

## Study selection

Titles of the articles identified through the search of the studies of influenza vaccination in the general population were first prescreened by one reviewer (SH). Abstracts and titles of articles meeting the prescreening criteria and of all articles identified through the liver disease-specific search were then screened by two independent reviewers (CP and SH). Studies deemed eligible were included in the full-text review by two independent reviewers (CP and SH). Disagreements were resolved by discussion and the reasons for exclusion were recorded. In case of uncertainty over eligibility, study authors were contacted. Studies for which this uncertainty could not be resolved were excluded (online supplementary table 5).

## Data collection process and data items

Using standardised online forms, two review authors (CP and SH) independently and in duplicate extracted data on: study participants (inclusion and exclusion criteria, method of recruitment/selection, study population characteristics), interventions and comparators (vaccine type, comparison treatment, dose, route of delivery, number and timing of vaccinations/comparator treatments, number of individuals and follow-up time in intervention and comparison groups), outcomes (definition, time points measured and reported, unit of measurement, number of outcomes in the intervention and control group, unadjusted and adjusted effect measures, covariates that the effect measures were adjusted for, comparisons, missing data and reasons for missingness, statistical methods used and processes for randomisation), study design (study type, country and setting, date of study, study duration, aim of study and withdrawals), study quality, study bias, study funding and conflicts of interest. We contacted study authors to obtain subgroup data, missing data and to clarify unclear data. Effect measures were collected in the format in which they were reported and transformed for analysis as required.

## Risk of bias in individual studies

Two review authors (CP and SH) independently assessed the risk of bias. We used the Cochrane Collaborations tool[20] to assess randomised trials and the Newcastle-Ottawa Scale[21] to assess observational studies. Age, sex, severity and aetiology of liver disease and presence of comorbidities were considered the most important confounders in the assessment of observational studies. Additionally, in studies of pandemic vaccine effect on clinical outcomes, we considered as a potential confounder a previous seasonal influenza vaccination in the same season the study vaccine was administered. Risk of bias assessment in serological studies focused on the study population with liver disease. Disagreements over risk of bias were resolved by discussion between reviewers.

## Summary measures and synthesis of results

Where more than a single study per outcome was identified and the study designs, protocols and measures of treatment effect were considered similar enough to produce a meaningful pooled effect, we used random-effects meta-analysis to summarise the average effects of vaccination. For serological vaccine response, we compared the haemagglutinisation inhibition (HI) antibody responses of patients with liver disease before and after vaccination. Comparisons between titres and titre ratios were presented as a mean difference (MD) of log geometric mean titre (GMT) or log geometric mean titre ratio (GMTR) with 95% CIs (titres and ratios were log transformed to back to their original scale to obtain SD). Seroconversion (defined as the proportion of patients whose negative prevaccination serum converted to an HI titre >1:40, or who experienced at least a fourfold titre increase, after vaccination) and seroprotection (defined as the proportion of patients achieving an HI titre of >1:40 after vaccination) levels were presented as prevalence rates (%) with 95% CI and comparisons between rates were presented as risk ratio (RR) with 95% CI. As reference levels, for a theoretically adequate response, we used 40% for seroconversion and 70% for seroprotection. Reported rates expressed only as percentages that did not correspond to an integer number of patients were imputed to the closest possible number of patients (closest integer above the reported non-integer for baseline and closest integer below for effect rates). Response measures were presented in categories by virus subtype. The vaccine effects on hospitalisation and mortality were compared between vaccinated and unvaccinated patients with liver disease and presented as a crude RR with 95% CI (adjusted effect measures, reported separately within results, were available from two studies but on different scales). Effect measures from multiple corresponding seasons were combined into an overall measure within each study. Presence of frailty bias were evaluated by contrasting the effect estimates in and out of influenza season. Person-time was presented as person-seasons for seasonal and as person-years for whole-year effect estimates. We regarded heterogeneity between studies as substantial if $\tau^2$ was >0, $I^2$ was >30% and the p value for Q-statistic was <0.10. Uncertainty in the estimates of between-study variance, likely to be greater when only a

small number of effect estimates are pooled together, was not quantified (option not available in the software used). Statistical analyses were performed using Stata V.14.0 and RevMan V.5.3.

## Additional analyses

Subgroup analyses and meta-regression were not carried out due to the low number of studies. Sensitivity analyses were carried out to investigate whether the serological vaccine effects were similar in less severe (non-cirrhotic disease) and advanced liver disease (cirrhotic disease) comparing response in all patients with that in patients with cirrhosis only. We were not able to conduct sensitivity analyses based on aetiology of disease as the study populations were mainly of mixed aetiologies. In studies that included a healthy control population, we compared antibody responses between patients with liver disease and the controls.

## Risk of bias across the studies

For each of the review outcomes, we evaluated the risk of selective outcome reporting bias using the Outcome Reporting Bias in Trials classification system.[22] Presence of publication bias was assessed based on this evaluation and our review design (funnel plots or formal tests were not used because of the low number of studies). The Grading of Recommendations Assessment, Development and Evaluation (GRADE) Working Group system[23] was used to assess and report the overall quality of evidence. In this assessment, we regarded a vaccine effect outside of the influenza season (suggesting the effect of frailty bias) for clinical outcomes as confounding (contributing to study limitations). All assessments were completed by two independent reviewers (CP and SH) and disagreements were resolved by discussion between reviewers.

## Patient and public involvement

Patients and the public were not involved in the design or planning of this study.

## RESULTS

The search of influenza studies in the general population provided a total of 10 041 unique study records. After title screening, 1607 of these records and 608 unique records from the liver disease-specific search were included in the abstract and title screening. The full text of 286 studies was examined in detail, and of these, 10 studies met the inclusion criteria for the review. An additional two studies were identified through recent influenza vaccine guidelines, making the total number of included studies 12. The study flow chart is presented in figure 1.

The included studies consisted of 1 randomised controlled trial and 11 cohort studies. Six studies included serological outcomes (table 1) and six included clinical outcomes (table 2). The studies with serological outcomes had low (four studies) to moderate bias (two

studies) in selection of study population and low (two studies), moderate (three studies) and high bias (one study) in assessment of the outcomes (bias mostly due to lack of description of independent, blind outcome assessment). Individual studies with clinical outcomes had moderate-to-low risk of bias in selection of the study population (in two studies vaccination status was self-reported) and low risk of bias in outcome assessment (majority used medical records). In most studies, the presence of confounding due to demographic and clinical characteristics was unclear (we did not have characteristics for the liver patient subgroups obtained from the studies conducted in the general population) and the results for control periods (non-influenza periods) were available only from two studies. High risk of selective outcome reporting was suspected in only one clinical study (outcome all-cause mortality) but mainly in serological studies (outcome GMTR). Evidence for all outcomes was judged as very low quality. Risk of bias and selective outcome reporting assessments, and the GRADE evidence profile are provided in online supplementary tables 6, 7 and 8.

The serological response to influenza vaccine was assessed in six studies including a total of 262 patients with liver disease (table 1). Each study employed a cohort design in which all study participants were vaccinated. Mean age of participants with liver disease ranged between 42 and 65 years (one study did not report baseline characteristics of participants). Three studies enrolled exclusively patients with cirrhosis in their liver disease group, two studies enrolled both patients with and without cirrhosis and one study had separate groups for patients with and without cirrhosis. Hepatitis C was the main cause of liver disease in three studies and hepatitis B in two studies. In one study, aetiology was not specified. Four studies immunised the participants with a trivalent seasonal vaccine and two used a monovalent pandemic vaccine. In all studies, blood samples for postvaccination measurements were drawn after minimum of 3 weeks and antibody titres measured using a traditional HI assay (antibody titre data are provided in online supplementary table 9).

HI antibody levels of patients with liver disease increased in response to vaccination (pooled random-effects MD of log GMTs 1.92, 95% CI 1.65 to 2.19) (figure 2). There was no substantial heterogeneity between individual estimates ($\tau^2$=0.04, $I^2$=15%, p=0.28) or levels against influenza A and B viruses (p=0.16). Seroconversion rate in patients with liver disease to both A/H1N1 and B subtypes was clearly above the 40% reference level (pooled random-effects prevalence for A/H1N1 79.95%, 95% CI 68.19% to 89.70% and pooled random-effects prevalence for B 86.82%, 95% CI 71.87% to 97.07%) and around 40% in the case of subtype A/H3N2 (pooled random-effects prevalence 55.53%, 95% CI 39.92% to 70.63%). While the majority of the studies reported estimates above 40%, there was substantial heterogeneity between response rates ($I^2$=76.55% and p<0.01 for subtype A/H1N1;

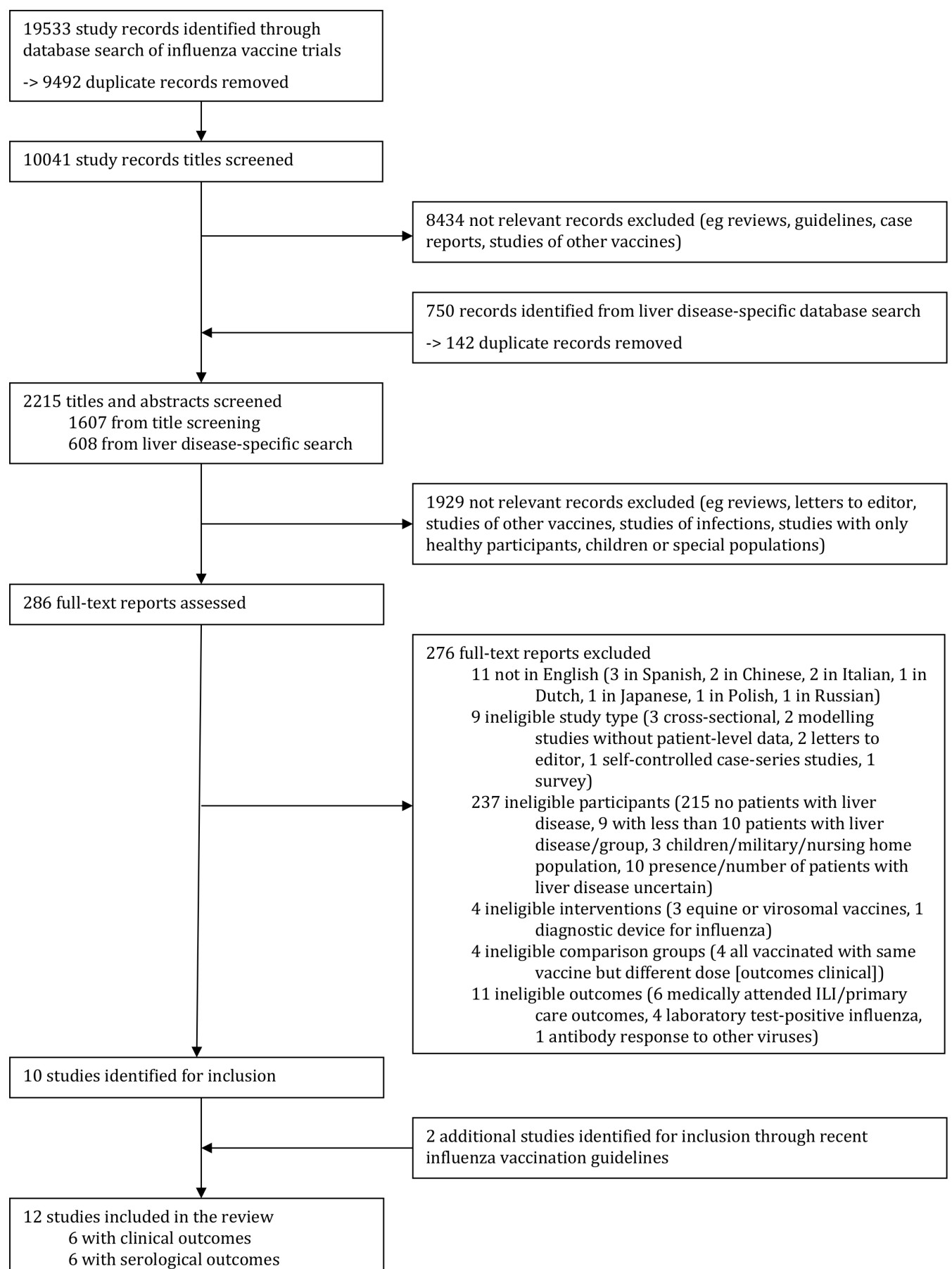

**Figure 1**  Study selection. ILI, influenza-like illness.

**Table 1** Characteristics of the included studies on serological response to influenza vaccine

| Study | Location | Design | Participants with liver disease (% cirrhosis, % viral aetiology) | Sample size | Vaccine | Follow-up (weeks) | Outcomes |
|---|---|---|---|---|---|---|---|
| Cheong et al[31] | South Korea | Cohort study with comparison groups | Patients with advanced cirrhosis (100% cirrhosis, 72% viral aetiology) | 50 (54 healthy controls) | Trivalent, split virus | 4 | GMT Seroconversion rate Seroprotection rate |
| Duchini et al[32] | USA | Cohort study with comparison groups | Patients with cirrhosis (100% cirrhosis, 56% viral aetiology) | 14 (9 healthy controls) | Trivalent, split virus | 6 | GMT GMTR Seroconversion rate |
| Hernández-Guerra et al[33] | Canary Islands, Spain | Cohort study with comparison groups | Patients with chronic HCV (0% cirrhosis, 100% viral aetiology) | 25 (15 healthy controls) | Monovalent, split virus | 4–11 | GMT GMTR Seroconversion rate Seroprotection rate |
| Ohfuji et al[34] | Japan | Cohort study | Patients with chronic HCV (25% cirrhosis, 100% viral aetiology) | 79 | Monovalent, split virus | 3 | GMT GMTR Seroconversion rate Seroprotection rate |
| Sayyad et al[35] | Iran | Cohort study with comparison groups | Patients with cirrhosis and inactive HBV carriers (47% cirrhosis, 100% viral aetiology in non-cirrhotics, aetiologies not provided in cirrhotics) | 28, 31 (34 healthy controls) | Trivalent, subunit | 4 | GMT Seroconversion rate Seroprotection rate |
| Soesman et al[36] | The Netherlands | Cohort study with comparison groups | Patients with cirrhosis (100% cirrhosis, aetiologies not provided) | 36 (45 healthy controls) | Trivalent, subunit | 4 | GMT Seroconversion rate Seroprotection rate |

GMT, geometric mean titre; GMTR, geometric mean titre ratio; HBV, hepatitis B virus; HCV, hepatitis C virus.

$I^2$=72.37% and p=0.01 for subtype A/H3N2; $I^2$=79.99% and p<0.01 for subtype B). Seroconversion rate is presented in figure 3. Seroprotection rate in patients with liver disease was above the 70% reference level (pooled random-effects rate 93.88%, 95% CI 87.33% to 98.40%) (figure 4). While the majority of studies reported a seroprotection rate above 70%, there was substantial heterogeneity ($I^2$=83.81%, p<0.01) (figure 4). Although some studies reported a substantial seroprotection level before vaccination, the pooled prevaccination seroprotection rate was clearly below 70% (online supplementary figure 1).

The effectiveness of influenza vaccine in preventing all-cause hospitalisation in patients with liver disease was investigated in two cohort studies, published in 2014 and 2016.[24 25] A study of 408 participants from Japan[24] investigated pandemic vaccine (against a single virus subtype) and enrolled exclusively patients with chronic hepatitis C virus (HCV) infection. A study of 8080 participants from

Taiwan[25] investigated seasonal vaccine (against multiple virus subtypes) exclusively in patients with chronic hepatitis B virus (HBV) infection. In the study of patients with HCV infection, patients were prospectively recruited from those who were under clinical follow-up for their liver disease but vaccination status was self-reported.[24] In the study of patients with HBV infection, both liver disease and vaccination status were determined from electronic medical claim records.[25] One study presented results for one influenza season and the other for an entire year after vaccination and pooled the results from a period of 9 years (figure 5). Both studies compared the rate of all-cause hospital admission (first incidence) between vaccinated and unvaccinated individuals.

Vaccinated patients with HCV were no less likely to be hospitalised than unvaccinated patients during an influenza season after pandemic influenza A vaccination (RR 0.57, 95% CI 0.24 to 1.37, risk difference [RD] −0.03, 95% CI −0.08 to 0.01).[24] In this study, hospitalisation

**Table 2** Characteristics of the included studies on clinical outcomes after influenza vaccination

| Study | Location | Design | Participants with liver disease | Sample size (patients with liver disease) | Study period | Outcomes |
|---|---|---|---|---|---|---|
| Campitelli et al[26] | Canada | Cohort study | Patients with chronic liver disease among the community-dwelling aged ≥65 years | 205 vaccinated/124 unvaccinated | Eight influenza seasons 1996–1998 and 2001–2007 | All-cause mortality. Acute respiratory illness-related hospitalisation (pneumonia and influenza). |
| Castilla et al[27] | Spain | Cohort study | Patients with cirrhosis among the community-dwelling aged ≥65 years | 3126 person-seasons vaccinated/1804 person-seasons unvaccinated Number of patients not reported | Two influenza seasons 2011–2013 | All-cause mortality. |
| Ohfuji et al[24] | Japan | Cohort study | Hospital outpatients with chronic HCV (64%, aged ≥65 years) | 132 vaccinated/276 unvaccinated | One influenza season | All-cause hospitalisation. |
| Song et al[29] | South Korea | Randomised controlled trial | Hospital inpatients and outpatients with cirrhosis (50%, aged ≥55 years) | 175 vaccinated/90 unvaccinated | One influenza season 2004–2005 | Influenza illness-related mortality (laboratory-confirmed influenza). |
| Su et al[25] | Taiwan | Cohort study | Hospital outpatients with chronic HBV (38%, aged ≥60 years) | 4434 vaccinated/3646 unvaccinated | Nine influenza seasons 2000–2009 | All-cause hospitalisation. All-cause mortality. Liver disease complication-related hospitalisation (septicaemia, bacteraemia, viraemia). |
| Vila-Córcoles et al[28] | Spain | Cohort study | Patients with cirrhosis among the community-dwelling (aged ≥65 years) | 100–117 vaccinated/73–104 unvaccinated (numbers varied per year) | Four influenza seasons, three non-influenza periods, three full years 2002–2005 | All-cause mortality. |

HBV, hepatitis B virus; HCV, hepatitis C virus.

outside the influenza season was not measured. Adjusting the effect estimate (for chronic comorbidities, steroid treatment within the preceding 6 months and albumin level), calculated in the study as the OR (crude OR 0.55, 95% CI 0.22 to 1.39), adjusted OR 0.43, 95% CI 0.16 to 1.17), did not reveal a significant effect.[24] Seasonal vaccination, however, was associated with a drop from 205/1000 patients to 149/1000 patients being hospitalised during a full year of follow-up (RD −0.06, 95% CI −0.07 to −0.04). Vaccinated patients with HBV were 27% less likely to be admitted to the hospital than unvaccinated patients (RR 0.73, 95% CI 0.66 to 0.80).[25] Both the crude and the adjusted time-to-event estimate that accounted for cirrhosis status, age, sex, pneumococcal vaccine, comorbidities and recent hospitalisation, calculated in the study as the HR, showed a protective effect

(crude HR 0.68, 95% CI 0.62 to 0.76, adjusted HR 0.56, 95% CI 0.50 to 0.62).[25] Based on a Kaplan-Meier plot, the difference in event-free survival probability between vaccinated and unvaccinated was established during the influenza season, approximately within the first 3 months after vaccination.[25] The effect of influenza vaccine on cause-specific hospitalisation was investigated in two studies.[25 26] A cohort study from Canada, published in 2010, measured acute respiratory illness-related hospitalisation during influenza season (defined as influenza/pneumonia hospitalisation) using medical claims and hospital records.[26] A total of 329 patients with liver disease were included as part of a general population, however, as there were five or fewer hospitalisation events in both groups, the study authors were unable to release the results (reason: institution's data policy, personal

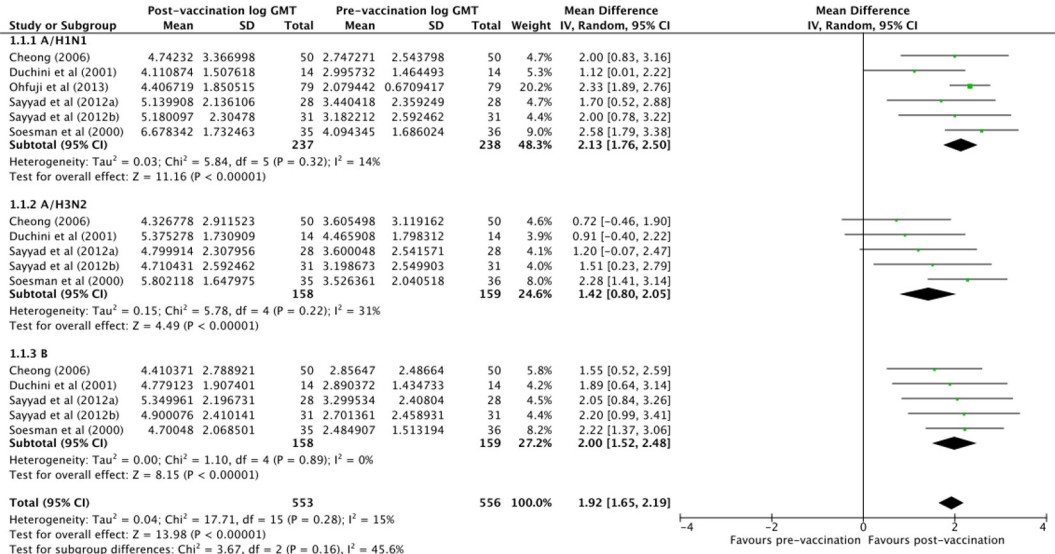

**Figure 2** Serological response to influenza vaccination: mean difference in log haemagglutinisation inhibition (HI) antibody geometric mean titres (GMTs) before and after vaccination. Sayyad *et al* (2012a) includes patients with cirrhosis and Sayyad *et al* (2012b) includes patients without cirrhosis.

communication). A liver disease complication-related hospitalisation outcome was included in the Taiwanese study of patients with HBV.[25] Vaccinated patients with liver disease were 57% less likely to be admitted to hospital for unspecified septicaemia, bacteraemia and viraemia than unvaccinated patients (RR 0.43, 95% CI 0.31 to 0.59). Vaccination was associated with a fall from 26/1000 patients to 11/1000 patients being hospitalised (RD −0.02, 95% CI −0.02 to −0.01). It appeared, however, that a difference in event-free survival probability was not

established during the influenza season (Kaplan-Meier plot).[25] No studies with the outcome influenza illness/ILI-related hospitalisation were identified.

The effectiveness of influenza vaccine in preventing all-cause mortality in patients with liver disease was investigated in five cohort studies,[25–28] published between 2007 and 2016. Only the Taiwanese study of patients with HBV enrolled exclusively patients with liver disease.[25] Two studies from Spain[27 28] and one from Canada[26] enrolled patients with liver disease as part of the general, community-dwelling population. All studies ascertained liver disease diagnosis and vaccination status from electronic

**Figure 3** Serological response to influenza vaccination: seroconversion rate after vaccination. Sayyad *et al* (2012a) includes patients with cirrhosis and Sayyad *et al* (2012b) includes patients without cirrhosis. ES=effect size.

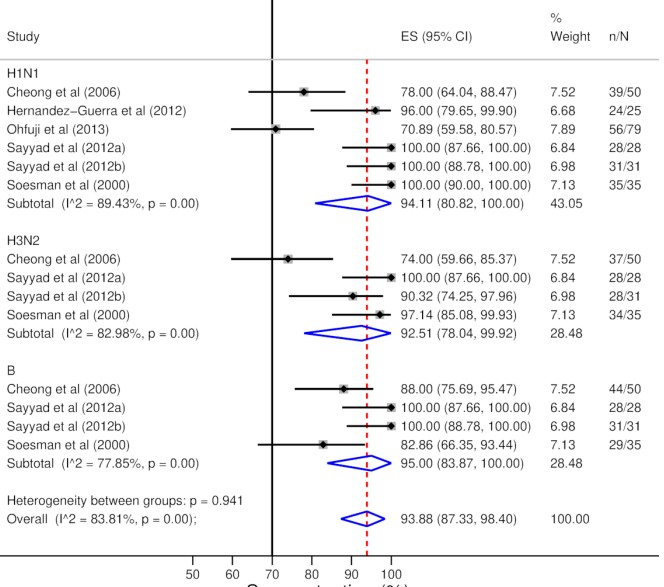

**Figure 4** Serological response to influenza vaccination: seroprotection rate after vaccination. Sayyad *et al* (2012a) includes patients with cirrhosis and Sayyad *et al* (2012b) includes patients without cirrhosis. ES=effect size.

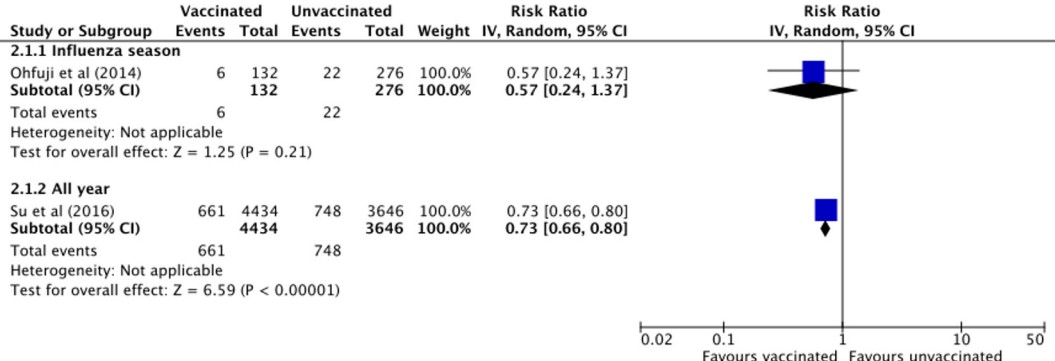

**Figure 5** Effect of influenza vaccine on clinical outcomes: all-cause hospitalisation in vaccinated compared with unvaccinated patients with liver disease. Ohfuji *et al*[24] investigated the effect of pandemic (monovalent) vaccine and SU *et al*[25] investigated the effect of seasonal vaccine (against three or more virus subtypes).

health or medical claim records. In one study, the record-based vaccination status was based on self-reporting. The Spanish studies followed patients over multiple seasons from the start of the study allowing them to contribute to both the unvaccinated and vaccinated follow-up time for different seasons,[27 28] whereas the other two studies followed patients up for a single season/year,[25 26] although this was over multiple seasons/years. All studies compared all-cause mortality rates between vaccinated and unvaccinated individuals.

Results are presented in figure 6. Vaccinated patients were no less likely to die during influenza season (pooled random effects RR 0.80, 95% CI 0.43 to 1.50, RD −0.001, 95% CI −0.01 to 0.01), or the entire year (pooled random effects RR 0.41, 95% CI 0.11 to 1.52, RD −0.06, 95% CI −0.08 to 0.04). All the studies that contributed to the influenza season-specific estimate reported non-significant effects and there was no substantial heterogeneity ($\tau^2$=0.12, $I^2$=33.00%, p=0.23). The studies that contributed to the whole-year estimate were substantially heterogeneous ($\tau^2$=0.83, $I^2$=94.00%, p<0.00), with the smaller Spanish study[28] reporting a non-significant effect (personal communication) and the larger Taiwanese study reporting a large protective effect (RR 0.22, 95% CI 0.17 to 0.28).[25] The Spanish study reported no effect outside the influenza period (RR 0.52, 95% CI 0.15 to 1.83). The Canadian study also measured the effect outside the non-influenza period but as there were five or fewer deaths in both groups, the authors were unable to provide the results (institution's data policy,

personal communication). Based on a Kaplan-Meier plot presented in the Taiwanese study, the difference in survival probability was mainly established approximately in the first 5 months after vaccination,[25] however, it was not clear whether the increase in death rate fully levelled off after the influenza season and it may be that this to some extent contributed to the size of the yearly effect estimate in this study. The effect of influenza vaccine on the rate of cause-specific mortality was investigated in one study.[29] The effect of influenza vaccine in preventing influenza illness/ILI-related mortality was included as an outcome in a South Korean randomised controlled trial of patients with cirrhosis published in 2007.[29] No deaths due to laboratory-confirmed influenza occurred among the 175 vaccinated or the 90 unvaccinated patients during the influenza season.[29] No studies investigating mortality related to acute respiratory illness or liver disease complications were identified.

To understand whether the vaccine effects were different based on the patients' cirrhosis status, we conducted sensitivity analyses. The serological response in patients with cirrhosis did not differ from the results of our main analyses (online supplementary figures 2-5). We did not have the data and therefore were not able to conduct similar analyses for the clinical outcomes.

In addition to comparing the vaccine effects within patients with liver disease, the studies of serological response (5/6 studies) also compared the outcomes between patients with liver disease and healthy controls (individuals without liver disease). The antibody titres

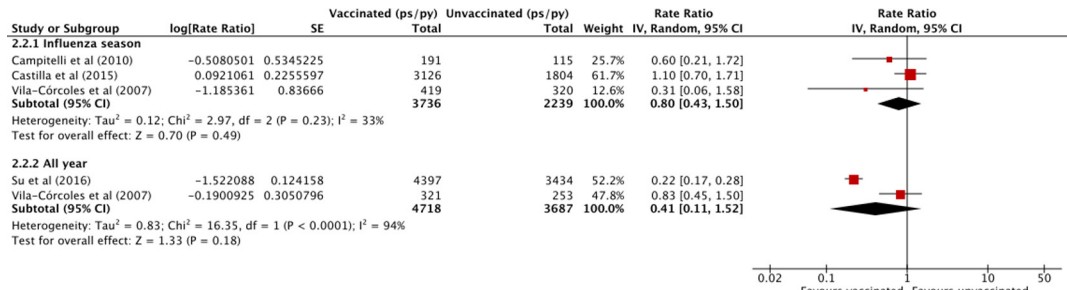

**Figure 6** Effect of influenza vaccine on clinical outcomes: all-cause mortality in vaccinated compared with unvaccinated patients with liver disease.

before or after vaccination, seroconversion rate and seroprotection rate of patients with liver disease did not differ from those of healthy individuals. Prevaccination to postvaccination titre ratios (GMTRs) reported were similar between patients with liver disease and healthy individuals except for influenza subtype A/H3N2 where the mean fold-increase in antibody levels of patients with liver disease was lower than that of healthy individuals. The pooled results for these comparisons are provided in online supplementary figures 6-11. The studies on clinical outcomes in patients with liver disease did not include healthy comparison groups and while we did not seek these data from the studies that included patients with liver disease as part of a general population, the all-cause mortality results in the entire general population can be found in online supplementary table 10.

## DISCUSSION

Six cohort studies from three continents (Asia, Europe and North America) investigated the serological vaccine response in patients with liver disease. Two cohort studies and one randomised controlled trial, all from Asia, investigated the vaccine effect on hospitalisation and/or mortality in patients with liver disease

exclusively. In addition, there are at least three cohort studies, two from Europe and one from North America, that have assessed the vaccine effect on these outcomes in a general community-dwelling elderly population including a subgroup with liver disease. Overall, the available evidence suggests that while influenza vaccine may not protect against all-cause mortality, it triggers an effective antibody response and may reduce the risk of all-cause hospitalisation in patients with liver disease (findings for clinical outcomes summarised in table 3, maximum quality obtainable is 'low' as all included studies are non-randomised).

A previous systematic review[14] on the effectiveness of inactivated influenza vaccine in different at-risk target groups identified only one randomised study in patients with liver disease. This study of 311 patients with cirrhosis, also included in our current review, found seasonal influenza vaccine effective against laboratory-confirmed influenza but not against ILI.[29] The study included hospitalised patients and could not investigate vaccine effectiveness against hospitalisation. Vaccine effect on influenza-related mortality was investigated, however, no deaths occurred during the study. It is possible that the effects we found are not entirely due

**Table 3** Summary of findings for clinical outcomes

Patients: adults with chronic liver disease
Setting: general population
Intervention: influenza vaccination (injectable, inactivated)
Comparison: no vaccination

| Outcome | Risk/rate in unvaccinated* | Risk/rate in vaccinated | Relative effect | Vaccine effectiveness | Number of participants/person-time (number and type of studies) | Quality of evidence (GRADE) |
|---|---|---|---|---|---|---|
| All-cause hospitalisation, influenza season (pandemic vaccine) | 80 per 1000 | 46 per 1000 (19–110) | 0.57 (0.24, 1.37) | Not significant | 408 (1 observational study) | Very low† |
| All-cause hospitalisation, all year | 205 per 1000 | 150 per 1000 (135–164) | 0.73 (0.66, 0.80) | 27% (20%, 34%)‡ | 8080 (1 observational study) | Very low§ |
| All-cause mortality, influenza season | 17 per 1000 person-seasons | 14 per 1000 person-seasons (7–26) | 0.80 (0.43, 1.50) | Not significant | 5975 person-seasons (3 observational studies) | Very low¶ |
| All-cause mortality, all year | 85 per 1000 person-seasons | 35 per 1000 person-seasons (9–129) | 0.41 (0.11, 1.52) | Not significant | 8405 person-years (2 observational studies) | Very low** |

Effectiveness of influenza vaccination in preventing all-cause hospitalisation and all-cause mortality in adults with chronic liver disease.
*Median risk in unvaccinated across the studies.
†Rated down from low to very low based on study limitations, imprecision of the estimate and suspicion of publication bias.
‡Could be applicable to patients with HBV only as the study only enrolled patients with HBV.
§Rated down from low to very low based on suspicion of publication bias.
¶Rated down from low to very low based on study limitations, inconsistency in absolute rates, imprecision of the estimate and suspicion of publication bias.
**Rated down from low to very low based on study limitations, imprecision of the estimate and suspicion of publication bias.
GRADE, Grading of Recommendations Assessment, Development and Evaluation; HBV, hepatitis B virus.

to the vaccine. Other factors that may have affected the clinical outcomes include frailty bias, competing risks (in case of our secondary, cause-specific outcomes) and differences in previous vaccinations, social support, healthcare system, lifestyle and in the severity or type of liver disease between the vaccinated and unvaccinated populations. Serological outcomes may have been further affected by variation of the antibody assays.

The data presented are predominantly from patients with viral liver disease. We cannot therefore be certain that our findings are applicable to patients with liver disease caused by alcohol, obesity or other non-viral factors. While sensitivity analyses showed that compared with non-cirrhotic liver disease, having cirrhosis did not worsen the antibody response to the vaccine, most data in the analyses of clinical outcomes come from patients with a milder form of liver disease than cirrhosis. We might expect that in severe liver disease complications from influenza are more frequent and we may have seen more outcome events if more patients with cirrhosis were included. Given the proxy effectiveness in patients with cirrhosis indicated by the antibody responses, this would mean that our study may have somewhat underestimated the true protective effect against clinical outcomes. However, all our data on the hospitalisation outcome are from high-income Asian countries and it is possible that the protective vaccine effect may not extend to populations from other geographical and income settings with different prevalence of comorbidities, age distribution, influenza seasonality and vaccine formulation used.

Our review has several strengths. It was conducted in a transparent way following established guidelines and a prospectively specified protocol. The search was conducted over multiple databases and we searched for both studies in patients with liver disease and studies in which patients with liver disease were included as part of the general population. To reduce the risk of missing relevant studies and relevant data and to increase the objectivity of the review, two independent reviewers screened the studies, extracted data and assessed the risk of bias. We studied all-cause outcomes to take into account the complex interplay of a chronic condition and acute infections. An underlying diagnosis of chronic liver disease may influence how cause of death and hospitalisation is recorded. This potentially reduces the number of cases that are attributed to influenza and solely investigating diagnoses of influenza is likely to substantially underestimate the impact of the vaccine.

There are also limitations to our review. The meta-analyses combine data from studies that differ in terms of patient population, outcome definitions and vaccine used. For the studies that included patients with liver disease as part of the general population, we had very limited information about the baseline characteristics of the patient with liver disease subgroup. In addition, despite the comprehensive search, publication bias, restriction on publication language and varying response to queries about data on liver disease subgroups may have affected the effects observed. We were not able to employ translation services and may have omitted relevant studies published in languages other than English from countries such as China where influenza vaccines are manufactured and liver disease is a significant public health challenge. A number of studies potentially included patients with liver disease as part of a wider study population but were unable to obtain confirmation and liver disease subgroup data from the study authors. Finally, the body of evidence identified in this review is limited both in quantity and quality, leaving room for uncertainty around the effects of the vaccination. The low quality is partly due to the non-randomised design of the included studies. This limitation may persist also in future reviews as conducting randomised controlled trials in a high-risk population are ethically challenging.

Evidence from mainly patients with viral liver disease suggests that underlying liver disease does not prevent patients from mounting an antibody response to influenza vaccine and similar proxy effectiveness is seen both in patients with and without cirrhosis. We also identified a protective influenza vaccine effect against yearly all-cause hospitalisation in patients with viral liver disease. Assuming this effect—estimated in a large cohort study of over 8000 patients—is due to the vaccine, seasonal influenza vaccination of 14–24 patients with viral liver disease could prevent one hospitalisation per year. It is uncertain whether the estimated effects extend to patients with liver disease of other than viral origin. No protective effect against all-cause mortality was found.

Current uptake of seasonal influenza vaccination is low. A recent systematic review identified low perceived vaccine effectiveness, and low concern and perceived risk of severe influenza infection as the most common barriers for seasonal influenza vaccination in patients with chronic illness.[30] Ideally, the evidence for influenza vaccine effectiveness in liver disease would come from a large randomised vaccine trial in patients with liver disease. In the absence of this evidence, we must rely on proxy estimates for effectiveness from serological studies and evidence on clinical outcomes from observational studies. The best available evidence on the effectiveness of influenza vaccine in liver disease, summarised in this review, shows both proxy effectiveness and protection against all-cause hospitalisation. The limited quantity and quality of this evidence means that the protective vaccine effect may be uncertain. However, considering the high risk of serious health outcomes from influenza infection in patients with liver disease and the safety and low cost of vaccination, overall the potential benefits of seasonal vaccination both to patients and the healthcare systems are likely to outweigh the costs and risks associated with vaccination. Wider collection of adult vaccination data and studies focused on patients with liver disease, using large databases linking routinely collected data on vaccinations and health, may allow us to address the remaining uncertainty about the effects of influenza vaccination in liver disease.

 

**Contributors** The study was conceived by SH, LS, AO'B and AH. SH developed the eligibility criteria, search strategy, risk of bias assessment strategy and data extraction plan with guidance from LS, AO'B and AH. SH (trained in systematic methods of literature searching as part of her PhD studies) performed the literature searches and prescreening of titles. SH and CAP reviewed titles and abstracts, and full-text articles, extracted data and assessed study bias and quality. SH analysed the data, and wrote the manuscript to which all authors CAP, LS, AO'B and AH contributed. The corresponding author had full access to all the data in the study and had final responsibility for the decision to submit for publication.

**Funding** This work was supported by the UK Biotechnology and Biological Sciences Research Council grant (grant number BBSRC BB/M009513/1) to SH. The funder had no role in study design, data collection, analysis and interpretation of data, or writing of the report.

**Disclaimer** The corresponding author had full access to all the data in the study and had final responsibility for the decision to submit for publication.

**Competing interests** None declared.

**Patient consent for publication** Not required.

**Provenance and peer review** Not commissioned; externally peer reviewed.

**Data availability statement** All data relevant to the study are included in the article or uploaded as supplementary information.

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
