## [Reviewer comments · BMJ Open]

ARTICLE DETAILS

TITLE (PROVISIONAL)	The effectiveness of influenza vaccines in adults with chronic liver disease: a systematic review and meta-analysis
AUTHORS	Härmälä, Suvi; Parisinos, Constantinos; Shallcross, Laura; O'Brien, Alastair; Hayward, Andrew

VERSION 1 – REVIEW

REVIEWER	Francesca Fortunato Department of Medical and Surgical Sciences, University of Foggia, Italy
REVIEW RETURNED	19-May-2019

GENERAL COMMENTS	Comments to the Authors A systematic review and meta-analysis to provide a systematic synthesis of the available evidence from both observational studies and randomised trials on the effectiveness of influenza vaccines to prevent serious health outcomes, hospitalisation, and death, in adults with chronic liver disease were carried out. The manuscript was written well and in line with the objective. The methods are described sufficiently and the supplementary reporting is complete. Background Page 5, line 8: Please add the references Material and Methods Page 7, line 27: I would suggest you modify the titles: “Eligibility criteria and search strategy” Page 7, line 34: Please specify that the study eligibility criteria have been defined using the PICOS (population, intervention, comparator, outcome, and study) design approach (Reference: Stone PW. Popping the (PICO) question in research and evidence based practice. Appl Nurs Res 2002;15:197-8. doi:10.1053/apnr.2002.34181) Page 9, line 2: I would suggest you add the following title: “Information source” Page 9, lines 6, 10, 13, 31: Please specify the reviewer's initial (...) Page 9, line 9: I would suggest you add the following title: “Quality assessment” Page 9, lines 10-32: I would recommend you shift this section to page 12 row 5 Page 9, line 10: Please specify the reviewer's initial (...) Results: Page 17, line 45: Please add “Figure 4”
---

REVIEWER	Jesus Castilla Instituto de Salud Pública de Navarra - IdiSNA, CIBERESP, Spain
REVIEW RETURNED	23-May-2019

GENERAL COMMENTS	This revision and meta-analysis investigated the influenza vaccination effect in adults with chronic liver diseases. The study has three main parts to evaluate whether influenza vaccination triggered serological response and prevented hospitalisation and death. The manuscript is well written and the methodology seems right. The main limitation is the lack of studies included in the revision with influenza confirmed hospitalizations or deaths as study outcome. In recent years there is increasing evidence of the influenza vaccine effectiveness from test-negative case-control studies. This design has demonstrated to be less affected by frailty bias. I wonder if there are studies with chronic liver patients using this methodology. Specific comments Abstract Page 3, line 3. Since evidence is limited to a final conclusion I would suggest changing the sentence "Vaccination had no effect on all-cause or cause-specific mortality or cause-specific hospitalisation." by something like "Vaccination effect could not be demonstrated against all-cause or cause-specific mortality or cause-specific hospitalisation." or "No protective effect against all-cause or cause-specific mortality or cause-specific hospitalisation was found." Page 11, line 20. "The vaccine effects on hospitalisation and mortality were compared between vaccinated and unvaccinated liver disease patients and presented as a crude RR with 95% CI." It seems a limitation to do not use the adjusted RR, since the adjustment correct confusion bias.
---

REVIEWER	José A López-López University of Murcia, Spain
REVIEW RETURNED	19-Jun-2019

GENERAL COMMENTS	This is a very thorough study. I only have some minor comments: Data analysis:  - The start of this section states that "two review authors independently extracted data". Could the authors clarify whether some or all studies were data-extracted by both reviewers and, if affirmative, if any appraisal of the degree of agreement among reviewers was considered? - Random-effects meta-analysis may be a suitable option for most scenarios. However, when only a small number of effect sizes are combined, estimation of the between-study variance parameter can be very imprecise and therefore it is good practice to also perform fixed-effect meta-analysis and compare the results. - Effect size calculation: could the authors provide further details on how mean differences were computed (e.g. did those take advantage of the comparison groups available from most studies?).
---

	- Since one of the outcomes is mortality, competing risks are a potential issue that should be considered and discussed in the paper. Results - Could the authors report confidence intervals around tau² and I².
--	--

REVIEWER	Michael A Stoto, PhD Georgetown University
REVIEW RETURNED	25-Jun-2019

GENERAL COMMENTS	From a methods point of view, this is a very thorough and thoughtful analysis, following all of the recommended standards for systematic reviews. Searches are clearly described and well executed, and the same can be said for the meta-analyses. The limitations of the available literature are clearly and honestly described. The authors appropriately capitalize on studies comparing vaccine effects in and outside of the flu season to gauge the frailty effect (PDF p. 12, ll. 20-25). My only quibble is that in the GRADE framework this effect should be regarded as “Plausible confounding, which would reduce a demonstrated effect” rather than “Imprecision.” The evidence is thin, and this is clearly presented, especially in Table 3 and the accompanying text, where the GRADE quality of evidence is described as “very low” for all four outcomes. As a statistician/epidemiologist, and a non-clinician, I see how the authors get from this to their conclusion that “The best available evidence on the effectiveness of influenza vaccine in liver disease, summarised in this review, shows both proxy effectiveness and protection against all-cause hospitalization” (PDF p. 26, ll. 55-60). However, I would regard the recommendation that follows (“this review provides sufficient evidence for physicians and public health agencies to actively promote seasonal influenza vaccination of liver disease patients, a group at a high risk of serious outcomes from influenza infection”) as weak using GRADE standards (Guyatt et al., BMJ 2008;336;924-926). However, citing only the sufficiency of the evidence, but not its consistently “very low” quality, in the abstract and “strengths and limitations” section is, to my mind, misleading.
--

VERSION 1 – AUTHOR RESPONSE

Reviewer #1

Reviewer Name: Francesca Fortunato

Institution and Country: Department of Medical and Surgical Sciences, University of Foggia, Italy

Please state any competing interests or state ‘None declared’: None declared

A systematic review and meta-analysis to provide a systematic synthesis of the available evidence from both observational studies and randomised trials on the effectiveness of influenza vaccines to prevent serious health outcomes, hospitalisation, and death, in adults with chronic liver disease were carried out. The manuscript was written well and in line with the objective. The methods are described sufficiently and the supplementary reporting is complete.

Our response:

We thank the reviewer for the kind words and are grateful for the suggestions to improve our manuscript. We have attempted to address all points raised and act on the suggestions and where the approach we have chosen differs from the approach suggested by the reviewer, provide clear reasons for our choices (please see our point-by-point response below).

1. Background, Page 5, line 8: Please add the references

Our response:

We have added the following reference to the second sentence that was missing a reference: Asrani SK, Devarbhavi H, Eaton J, et al. Burden of liver diseases in the world. *J Hepatol* 2019;70:151–71. doi:10.1016/j.jhep.2018.09.014

The sentence now reads (page 5):

“The prevalence of liver disease is increasing, driven by obesity and alcohol consumption [2].”

2. Material and Methods, Page 7, line 27: I would suggest you modify the titles: “Eligibility criteria and search strategy”

Our response:

Based on the reviewer’s excellent suggestion to improve the subheadings to better describe the content of the separate Method sections, using the suggestions and the PRISMA guidelines as our guide, we have revised the subheadings throughout the Methods section.

Instead of the subheadings:

- Search strategy and selection criteria
- Data analysis
- Role of the funding source
- Patient and public involvement

The Methods section is now structured around the following subheadings (page 7-13):

- Protocol and registration (this section has been moved from the end of the Methods section to the beginning of the Methods section)
- Eligibility criteria
- Information sources and search strategy
- Study selection
- Data collection process and data items
- Risk of bias in individual studies
- Summary measures and synthesis of results (for better fit we also moved the sentence about the software used to perform the statistical analyses that was placed in the end of the paragraph describing the sensitivity/additional analyses to the end of this section)
- Additional analyses
- Risk of bias across the studies
- Role of the funding source
- Patient and public involvement.

3. Page 7, line 34: Please specify that the study eligibility criteria have been defined using the PICOS (population, intervention, comparator, outcome, and study) design approach (Reference: Stone PW. Popping the (PICO) question in research and evidence based practice. *Appl Nurs Res* 2002;15:197-8. doi:10.1053/apnr.2002.34181)

Our response:

As suggested, we have now added the specification that the eligibility criteria was defined using the PICOS approach.

Instead of:

“Studies published in peer-reviewed journals were eligible for inclusion for this systematic review and meta-analysis if they met the following criteria:...”

The text now reads (page 7):

“Studies published in peer-reviewed journals were eligible for inclusion for this systematic review and meta-analysis if they met the following PICOS (participants, intervention, comparator, outcome and study design) criteria:...”

4. Page 9, line 2: I would suggest you add the following title: “Information source”

Our response:

We hope that we have adequately acted on the reviewer’s suggestions to improve the flow of the methods section and addressed this point in our response to the comment number 2 above.

5. Page 9, lines 6, 10, 13, 31: Please specify the reviewer's initial (...)

Our response:

We thank the reviewer for noticing the missing initials. These have now been added to the text in brackets as follows:

Initials added on page 9:

“Titles of the articles identified through the search of the studies of influenza vaccination in the general population were first pre-screened by one reviewer (SH). Abstracts and titles of articles meeting the pre-screening criteria and of all articles identified through the liver disease-specific search were then screened by two independent reviewers (CP and SH). Studies deemed eligible were included in the full-text review by two independent reviewers (CP and SH).”

Further initials added on page 9:

“Using standardised online forms, two review authors (CP and SH) independently extracted data on:...”

6. Page 9, line 9: I would suggest you add the following title: “Quality assessment”

Our response:

We hope that we have adequately acted on the reviewers suggestions to improve the flow of the methods section and addressed this point in our response to the comment number 2 above.

7. Page 9, lines 10-32: I would recommend you shift this section to page 12 row 5

Our response:

As the first assessment is for risk of bias in individual studies and the second assessment that comes after the evidence synthesis is for risk of bias that may affect the cumulative evidence, we have decided to leave this section to its original location within the manuscript. We hope the revised subheaders within the Methods section has improved the flow of the text in general and sequence of the risk of bias assessments is now clearer to follow.

8. Page 9, line 10: Please specify the reviewer's initial (...)

Our response:

We again thank the reviewer for noticing the missing initials (we believe that here the reviewer has meant missing initials on page 10, line 10 as missing initials on page 9 were noted on a previous comment). These have now been added to the text in brackets as follows.

Initials added on page 10:

“Two review authors (CP and SH) independently assessed the risk of bias.”

We found another place within the text where reviewer initials were missing and have also added these to the text in brackets as follows.

Initials added on page 13:

“All assessments were completed by two independent reviewers (CP and SH) and disagreements were resolved by discussion between reviewers.”

9. Results, Page 17, line 45: Please add “Figure 4”

Our response:

We have added the missing reference to Figure 4 in brackets as follows.

Reference to figure 4 added on page 18:

“While the majority of studies reported a seroprotection rate above 70%, there was substantial heterogeneity ($I^2=83\cdot81\%$, $p<0.01$) (Figure 4).”

Reviewer #2

Reviewer Name: Jesus Castilla

Institution and Country: Instituto de Salud Pública de Navarra - IdiSNA, CIBERESP, Spain

Please state any competing interests or state ‘None declared’: None declared

This revision and meta-analysis investigated the influenza vaccination effect in adults with chronic liver diseases. The study has three main parts to evaluate whether influenza vaccination triggered serological response and prevented hospitalisation and death.

The manuscript is well written and the methodology seems right.

Our response:

We thank the reviewer for the kind words and are grateful for the suggestions to improve our manuscript. We have attempted to address all points raised and act on the suggestions and where the approach we have chosen differs from the approach suggested by the reviewer, provide clear reasons for our choices (please see our point-by-point response below).

1. The main limitation is the lack of studies included in the revision with influenza confirmed hospitalizations or deaths as study outcome. In recent years there is increasing evidence of the influenza vaccine effectiveness from test-negative case-control studies. This design has demonstrated to be less affected by frailty bias. I wonder if there are studies with chronic liver patients using this methodology.

Our response:

We did find some test-negative case-control studies that included or are likely to have included patients with liver disease. The patients in these studies were hospitalised (or the study included also hospitalised patients) for influenza-like-illness (ILI). Although only cases had laboratory-confirmed influenza, because our related secondary outcomes of interest were influenza illness/ILI-related hospitalisation, this would have meant that both the cases and controls would have had the outcome event and so these studies were excluded. To clarify the exclusion of these studies from the review, we have added a clarification to the Methods section.

Instead of:

“Review articles, case reports, cross-sectional studies, animal studies, editorials, clinical guidelines, studies with liver transplant patients only (response to the vaccination and clinical outcomes are likely to be strongly influenced by the immunosuppressive medication rather than the status of liver disease), studies focused on special populations (such as pregnant women, nursing home residents, or patients with other chronic diseases), and studies retracted from publication were excluded. “

The text now reads (page 8):

“Review articles, case reports, cross-sectional studies, animal studies, editorials, clinical guidelines, studies with liver transplant patients only (response to the vaccination and clinical outcomes are likely to be strongly influenced by the immunosuppressive medication rather than the status of liver disease), studies focused on special populations (such as pregnant women, nursing home residents, or patients with other chronic diseases), test-negative case-control studies (where both cases and controls were hospitalised for influenza-related illness/ILI/acute respiratory illness, listed in Supplementary materials, Table 2), and studies retracted from publication were excluded. “

To provide further details of these studies as they may be of interest to the readers and for future reviews looking at lab-confirmed influenza outcomes, we have also added the list of the test-negative studies that included or are likely to have included liver patients in the Supplementary materials ,Table 2 (page 3).

2. Abstract, Page 3, line 3. Since evidence is limited to a final conclusion I would suggest changing the sentence “Vaccination had no effect on all-cause or cause-specific mortality or cause-specific hospitalisation.” by something like “Vaccination effect could not be demonstrated against all-cause or cause-specific mortality or cause-specific hospitalisation.” or “No protective effect against all-cause or cause-specific mortality or cause-specific hospitalisation was found.”

Our response:

We thank the reviewer for suggesting an improved way of describing the result and accordingly have amended the sentence.

Instead of: “Vaccination had no effect on all-cause or cause-specific mortality or cause-specific hospitalisation.”

The text now reads (page 3): “No effect against all-cause or cause-specific mortality or cause-specific hospitalisation was found.”

3. Page 11, line 20. “The vaccine effects on hospitalisation and mortality were compared between vaccinated and unvaccinated liver disease patients and presented as a crude RR with 95% CI.” It seems a limitation to do not use the adjusted RR, since the adjustment correct confusion bias.

Our response:

The adjusted effect measures were available from 2 of the studies on clinical outcomes that studied the vaccine effect exclusively in liver patients, Ohfuji et al (2014) and Su et al (2016). Ohfuji et al (2014) reported their results as odds ratios without providing follow-up times for the comparison groups. Su et al (2016) reported hazard ratios, follow-up times for the comparison groups and the number of patients in each group (each patient was followed up to a year since flu vaccination, and the person-years in each group were similar to the number of patients in each group). To present the study results on a common scale for the hospitalisation outcome, we decided to present them as risk ratios. The crude effect estimates from the studies are similar but slightly more conservative than the adjusted estimates. We felt that the choice to use the crude estimates to summarise the results on a common and easily understandable scale was a sensible one.

For clarity, we reported both the crude and adjusted hazard ratios from Su et al (2016) within the text describing the results. We have now added also the crude and adjusted odds ratios from Ohfuji et al (2014) to the text.

Instead of:

“Adjusting the effect estimate (for chronic co-morbidities, steroid treatment within the preceding 6 months and albumin level), calculated in the study as the odds ratio, did not reveal a significant effect [31].”

The text about the results of Ohfuji et al (2014) now reads (page 19):

“Adjusting the effect estimate (for chronic co-morbidities, steroid treatment within the preceding 6 months and albumin level), calculated in the study as the odds ratio (crude odds ratio 0.55, 95% CI [0.22,1.39]), adjusted odds ratio 0.43, 95% CI [0.16,1.17]), did not reveal a significant effect [31].”

And to clarify the choice of using crude estimates, we have added the following clarification to the Methods section:

Instead of:

“The vaccine effects on hospitalisation and mortality were compared between vaccinated and unvaccinated liver disease patients and presented as a crude RR with 95% CI.”

The sentence about effect measures now reads (page 12):

“The vaccine effects on hospitalisation and mortality were compared between vaccinated and unvaccinated liver disease patients and presented as a crude RR with 95% CI (adjusted effect measures, reported separately within results, were available from two studies but on different scales).”

Reviewer #3

Reviewer Name: José A López-López

Institution and Country: University of Murcia, Spain

Please state any competing interests or state ‘None declared’: None declared

This is a very thorough study. I only have some minor comments.

Our response:

We thank the reviewer for the kind words and are grateful for the suggestions to improve our manuscript. We have attempted to address all points raised and act on the suggestions and where

the approach we have chosen differs from the approach suggested by the reviewer, provide clear reasons for our choices (please see our point-by-point response below).

1. Data analysis: The start of this section states that "two review authors independently extracted data". Could the authors clarify whether some or all studies were data-extracted by both reviewers and, if affirmative, if any appraisal of the degree of agreement among reviewers was considered?

Our response:

We thank the reviewer for pointing us to clarify this statement. To avoid missing data and errors in extracted data, two of us extracted data on each of the included studies on each of the specified data items. We did not assess the degree of agreement but used any differing extractions as a clue to double check the data in the original paper and to contact the study authors where the data was not clear or the data was deemed missing.

We have amended the corresponding sentence to clarify that the data extraction was conducted in duplicate.

Instead of: "Using standardised online forms, two review authors independently extracted data on..."

The sentence now reads (page 9): "Using standardised online forms, two review authors independently and in duplicate extracted data on..."

2. Random-effects meta-analysis may be a suitable option for most scenarios. However, when only a small number of effect sizes are combined, estimation of the between-study variance parameter can be very imprecise and therefore it is good practice to also perform fixed-effect meta-analysis and compare the results.

Our response:

We chose to perform random-effects meta-analysis as we felt it is a plausible assumption that the individual studies are not all conducted in an exactly the same way in an exactly same kind of population (i.e. the design, conduct and population included in the individual studies may have influenced the results) and the random-effects model would account for the possibility that each individual study is actually trying to estimate a slightly different effect. We thank the reviewer for pointing out the limitations of the random-effects method where substantial heterogeneity is clearly present and only a small number of effect estimates are pooled together, and for directing us to have a closer look at the results of our meta-analyses where this concern should be investigated. This is definitely the situation in our meta-analysis of all-cause mortality.

In this analysis, only 3 studies were pooled together to estimate the average effect of influenza vaccination on all-cause mortality during the flu season and only 2 studies were pooled together to estimate the average effect during the entire year. During the flu season, in the random-effect analysis the smaller studies that have a point estimate with a protective direction get a larger weight than in fixed-effects analysis and so the pooled point estimate is slightly more in the protective direction in the random-effects analysis (pooled RR 0.80, 95% CI [0.43, 1.50]) compared to the fixed-effects (pooled RR 0.93, 95% CI [0.63, 1.39]). The conclusions however are not affected by this small-study effect as neither analysis shows a pooled protective vaccine effect when the confidence intervals (only slightly wider in the random-effects analysis) are taken into account.

In the all-year analysis, the results differ greatly between the two analysis methods. In the random-effect analysis the smaller study with no vaccine effect is weighted nearly equally to the larger study with a strongly protective vaccine effect, the large difference in the effect estimates of these two studies results in wide confidence intervals and we find no effect on all-cause mortality on average

(pooled RR 0.41, 95% CI [0.11, 1.52]). In contrast, in the fixed-effects model, the larger study with protective effect accounts for more than 85% of the results and we find a strongly protective vaccine effect with narrow confidence intervals (RR 0.26, 95% CI [0.21, 1.33]). While our choice of analysis method clearly and greatly influences the result of this analysis, given the potential differences in methodology and the study population between the 2 included studies and the possibility of the frailty bias affecting the results of larger study, we feel the the random-effect analysis provide a more plausible and truthful representation of the effect of the vaccine on all-cause mortality and the uncertainty around it.

3. Effect size calculation: could the authors provide further details on how mean differences were computed (e.g. did those take advantage of the comparison groups available from most studies?).

Our response:

Our main interest, in terms of serological outcomes, was to better understand the liver patients' antibody response to the vaccination and so the mean differences were computed comparing the assay results in liver patients before vaccination and after the vaccination. In additional analyses, we also took advantage of the healthy comparison groups and compared the antibody responses between liver disease patients and the healthy. In these analyses, the mean differences were computed comparing the post-vaccination assay results in liver patients and healthy individuals. To provide context for the post-vaccination comparison between liver patients and healthy, we also computed the mean difference comparing the assay results before the vaccination. The results of these additional analyses with mean difference in antibody responses are provided in the Supplementary materials figures 6-8.

To further clarify how the mean difference was computed in the manuscript text, we have amended the corresponding text in the Methods section as follows.

Instead of: "For serological vaccine response, we compared haemagglutination inhibition (HI) antibody responses before and after vaccination."

This sentence now reads (page 11): "For serological vaccine response, we compared the liver patients' haemagglutination inhibition (HI) antibody responses before and after vaccination."

We have also amended the following sentence in the Methods section to make it clear that all patients in these analyses were vaccinated.

Instead of: "Seroconversion (defined as the negative pre-vaccination serum converting to a HI titre >1:40 or at least a four-fold titre increase) and seroprotection (defined as the proportion of vaccinated individuals achieving a HI titre of > 1:40) levels were presented..."

The sentence now reads (page 11): "Seroconversion (defined as the proportion of patients whose negative pre-vaccination serum converted to a HI titre >1:40, or who experienced at least a four-fold titre increase, after vaccination) and seroprotection (defined as the proportion of patients achieving a HI titre of > 1:40 after vaccination) levels were presented..."

4. Since one of the outcomes is mortality, competing risks are a potential issue that should be considered and discussed in the paper.

Our response:

We thank the reviewer for reminding us of the potential effect of competing risks in our analyses of the cause-specific outcomes. To acknowledge this, we have revised (and corrected a small typo in) the sentence on alternative explanations for the results in the Discussion section.

Instead of:

“Other factors that may have affected the clinical outcomes include frailty bias, previous vaccinations, social support, healthcare system, lifestyle, and in the severity or type of liver disease between the vaccinated and unvaccinated populations.”

The text now reads (page 24):

“Other factors that may have affected the clinical outcomes include frailty bias, competing risks (in case of our secondary, cause-specific outcomes) and differences in previous vaccinations, social support, healthcare system, lifestyle, and in the severity or type of liver disease between the vaccinated and unvaccinated populations.”

5. Results: Could the authors report confidence intervals around tau² and I².

Our response:

Unfortunately, we were not able to compute the confidence intervals around the between-study-variance estimates. To clarify this and acknowledge that there may be great uncertainty in the heterogeneity estimates when a small number of effect estimates are pooled together, we have added the following sentence to the methods section.

Sentence added on page 12:

“Uncertainty in the estimates of between-study variance, likely to be greater when only a small number of effect estimates are pooled together, was not quantified (option not available in the software used).”

Reviewer #4

Reviewer Name: Michael A Stoto, PhD

Institution and Country: Georgetown University

Please state any competing interests or state 'None declared': None declared.

From a methods point of view, this is a very thorough and thoughtful analysis, following all of the recommended standards for systematic reviews. Searches are clearly described and well executed, and the same can be said for the meta-analyses. The limitations of the available literature are clearly and honestly described.

Our response:

We thank the reviewer for these kind words and are grateful for and have attempted to act on all the suggestions to improve our protocol (please see our point-by-point response below).

1. The authors appropriately capitalize on studies comparing vaccine effects in and outside of the flu season to gauge the frailty effect (PDF p. 12, ll. 20-25). My only quibble is that in the GRADE framework this effect should be regarded as “Plausible confounding, which would reduce a demonstrated effect” rather than “Imprecision.”

Our response:

We thank the reviewer for noticing this. We had considered the frailty effect as part of the study limitations in our GRADE assessment (Supplementary materials Table 8, page 13) but mistakenly considered it also as a marker of imprecision.

To correct and clarify this in the manuscript, we have amended the sentence describing the GRADE assessment as follows.

Instead of:

“In this assessment, we considered a vaccine effect outside of the influenza season (suggesting the effect of frailty bias) for clinical outcomes as marker for imprecision.”

The sentence now reads (page 13):

“In this assessment, we regarded a vaccine effect outside of the influenza season (suggesting the effect of frailty bias) for clinical outcomes as confounding (contributing to study limitations).”

The results of our assessments did not change when correcting this error.

2. The evidence is thin, and this is clearly presented, especially in Table 3 and the accompanying text, where the GRADE quality of evidence is described as “very low” for all four outcomes. As a statistician/epidemiologist, and a non-clinician, I see how the authors get from this to their conclusion that “The best available evidence on the effectiveness of influenza vaccine in liver disease, summarised in this review, shows both proxy effectiveness and protection against all-cause hospitalization” (PDF p. 26, ll. 55-60). However, I would regard the recommendation that follows (“this review provides sufficient evidence for physicians and public health agencies to actively promote seasonal influenza vaccination of liver disease patients, a group at a high risk of serious outcomes from influenza infection”) as weak using GRADE standards (Guyatt et al., BMJ 2008;336;924-926). However, citing only the sufficiency of the evidence, but not its consistently “very low” quality, in the abstract and “strengths and limitations” section is, to my mind, misleading.

Our response:

We thank the reviewer for reminding us to be clearer on the limitations of the evidence found and directing us to re-think our conclusions and their consistency with the GRADE standards. Accordingly, we have attempted to improve the Abstract and the strengths and limitations paragraph and the final conclusions in the Discussion section to better reflect the uncertainty of the vaccine effect.

In the Abstract we have revised both the results and conclusions section (text changes that are not related to improving the clarity around uncertainty have been made to stay within the word limit).

Instead of:

“Results: Our searches provided 10649 unique records, 286 were eligible for full-text review and 12 studies were included. Majority of the patients had viral liver disease. Both non-cirrhotic and cirrhotic patients mounted an antibody response to the vaccine and the response was similar to that of healthy. Current evidence on clinical outcomes is limited but shows that seasonal influenza vaccination is associated with a reduction in risk of hospital admission from 205/1000 to 149/1000 (risk difference -0.06, 95% CI [-0.07, -0.04]) in patients with viral liver disease. Vaccinated patients were 27% less likely to be admitted to hospital compared to unvaccinated patients (risk ratio 0.73, 95% CI [0.66, 0.80]). Vaccination had no effect on all-cause or cause-specific mortality or cause-specific hospitalisation.

Conclusions: Influenza vaccination represents a safe, low-cost and readily available treatment to prevent frequent hospitalisations in liver disease but its uptake is poor. Our review provides sufficient

evidence of vaccine effectiveness for physicians and public health agencies to actively promote vaccination of liver patients, a group at a high risk of serious outcomes from influenza.”

These sections now read (page 2-3):

“Results: We found 10649 unique records, 286 were eligible for full-text review and 12 were included. Most patients had viral liver disease. All studies were of very low quality. Both non-cirrhotic and cirrhotic patients mounted an antibody response and influenza vaccination was associated with a reduction in risk of hospital admission from 205/1000 to 149/1000 (risk difference -0.06, 95% CI -0.07, -0.04) in patients with viral liver disease. Vaccinated patients were 27% less likely to be admitted to hospital compared to unvaccinated patients (risk ratio 0.73, 95% CI 0.66, 0.80). No effect against all-cause or cause-specific mortality or cause-specific hospitalisation was found.

Conclusions: The low quantity and quality of the evidence means that the protective vaccine effect may be uncertain. Considering the high risk of serious health outcomes from influenza infection in liver patients and the safety and low cost of vaccination, overall, the potential benefits of seasonal vaccination both to patients and the healthcare systems are likely to outweigh the undesirable consequences of severe influenza infection.”

In the Discussion section describing the strengths and limitations, we have revised the sentence about the quality of the evidence.

Instead of:

“Finally, the body of evidence identified in this review consists almost exclusively of studies with a non-randomised design. While this decreases the overall quality of evidence, conducting randomised controlled trials in a high-risk population are ethically challenging.”

The sentence now reads (page 26):

“Finally, the body of evidence identified in this review is limited both in quantity and quality, leaving room for uncertainty around the effects of the vaccination. The low quality is partly due to the non-randomised design of the included studies. This limitation may persist also in future reviews as conducting randomised controlled trials in a high-risk population is ethically challenging.”

Finally, in the Discussion section, we have revised our conclusions.

Instead of:

“The best available evidence on the effectiveness of influenza vaccine in liver disease, summarised in this review, shows both proxy effectiveness and protection against all-cause hospitalisation. Therefore, we believe, this review provides sufficient evidence for physicians and public health agencies to actively promote seasonal influenza vaccination of liver disease patients, a group at a high risk of serious outcomes from influenza infection.”

The text now reads (page 27):

“The best available evidence on the effectiveness of influenza vaccine in liver disease, summarised in this review, shows both proxy effectiveness and protection against all-cause hospitalization. The limited quantity and quality of this evidence means that the protective vaccine effect may be uncertain. However, considering the high risk of serious health outcomes from influenza infection in liver patients and the safety and low cost of vaccination, overall the potential benefits of seasonal vaccination both to patients and the healthcare systems are likely to outweigh the undesirable consequences of severe influenza infection. Wider collection of vaccination data in adults and liver patient -focused studies using large databases linking routinely collected data on vaccinations and health may allow us to address the remaining uncertainty about the effects of influenza vaccination in liver disease.”

VERSION 2 – REVIEW

REVIEWER	Francesca Fortunato University of Foggia, Foggia, Italy
REVIEW RETURNED	29-Jul-2019

GENERAL COMMENTS	The manuscript was revised. I have no further comments to add.
--

REVIEWER	José A López-López University of Murcia, Spain
REVIEW RETURNED	09-Aug-2019

GENERAL COMMENTS	The authors have adequately addressed my comments, and I am happy to recommend the publication of this paper.
---

REVIEWER	Michael A Stoto Georgetown University, U.S.A.
REVIEW RETURNED	25-Jul-2019

GENERAL COMMENTS	I have reviewed the revised version of the paper, along with the authors' responses to all of the reviewers, and find that the authors have been very responsive to all of our concerns. It seems, though, that the authors have inadvertently misstated the "down side" of vaccination in the following sentence in the abstract and discussion section. "Considering the high risk of serious health outcomes from influenza infection in liver patients and the safety and low cost of vaccination, overall, the potential benefits of seasonal vaccination both to patients and the healthcare systems are likely to outweigh the undesirable consequences of severe influenza infection." In particular, the sentence, should say that the benefits ... outweigh "the costs and risks associated with vaccination" rather than "the undesirable consequences of severe influenza infection." Avoiding the consequences of influenza is a benefit!
--

VERSION 2 – AUTHOR RESPONSE

Reviewer #4

Reviewer Name: Michael A Stoto

Institution and Country: Georgetown University, U.S.A.

Please state any competing interests or state 'None declared': None declared

1. I have reviewed the revised version of the paper, along with the authors' responses to all of the reviewers, and find that the authors have been very responsive to all of our concerns.

It seems, though, that the authors have inadvertently misstated the "down side" of vaccination in the following sentence in the abstract and discussion section.

"Considering the high risk of serious health outcomes from influenza infection in liver patients and the safety and low cost of vaccination, overall, the potential benefits of seasonal vaccination both to patients and the healthcare systems are likely to outweigh the undesirable consequences of severe influenza infection."

In particular, the sentence, should say that the benefits ... outweigh "the costs and risks associated with vaccination" rather than "the undesirable consequences of severe influenza infection." Avoiding the consequences of influenza is a benefit!

Our response:

We thank the reviewer for reviewing our revised manuscript and for noticing this unclear statement. As suggested, we have now revised the end of the sentence as follows.

Instead of:

“... the potential benefits of seasonal vaccination both to patients and the healthcare systems are likely to outweigh the undesirable consequences of severe influenza infection.”

The sentence now reads (in abstract, page 3, and in the Discussion section, page 27):

“... the potential benefits of seasonal vaccination both to patients and the healthcare systems are likely to outweigh the costs and risks associated with vaccination.”

VERSION 3 – REVIEW

REVIEWER	Michael Stoto Georgetown University, USA
REVIEW RETURNED	13-Aug-2019
GENERAL COMMENTS	The change in the abstract and conclusions fully addresses my earlier concern.